



# Sval_Imp_v1: A gridded forcing dataset for climate change impact research on Svalbard

Thomas Vikhamar Schuler[1] and Torbjørn Ims Østby[1,2]

[1]Department of Geosciences, University of Oslo, Norway
[2]now at: Lyse AS, Stavanger, Norway

**Correspondence:** Thomas V Schuler (t.v.schuler@geo.uio.no)

**Abstract.** We present Sval_Imp_v1, a high resolution gridded dataset designed for forcing models of terrestrial surface processes on Svalbard. The dataset is defined on a 1km grid covering the archipelago of Svalbard, located in the Norwegian Arctic (74-82°N). Using a hybrid methodology combining multi-dimensional interpolation with simple dynamical modelling, the atmospheric reanalyses ERA-40 and ERA-interim by the European Centre for Medium-Range Weather Forecasting have been
5 downscaled to cover the period 1957-2017 at steps of 6h. The dataset is publicly available from a data repository. In this paper, we describe the methodology used to construct the dataset, present the organization of the data in the repository and discuss the performance of the downscaling procedure. In doing so, the dataset is compared to a wealth of data available from operational as well as project-based measurements. The quality of the downscaled dataset is found to vary in space and time, but generally represents an improvement compared to unscaled values, especially for precipitation. Whereas operational records are biased to
10 low-elevations around the fringes of the archipelago, we stress the hitherto under-used potential of project-based measurements for evaluating atmospheric models. For instance, records of snow accumulation on large ice masses may represent measures of seasonally-integrated precipitation in regions sensitive to the downscaling procedure, thus providing added value.

Sval_Imp_v1 (Schuler, 2018) is publicly available from the Norwegian Research Data Archive NIRD, a data repository (https://doi.org/10.11582/2018.00006).

## 1 Introduction

The non-linearity of many surface processes poses challenges on appropriateness of atmospheric forcing for impact studies in terms of accuracy and precision (e.g., Liston and Elder, 2006). Especially in mountainous areas, the variability of surface
20 systems is typically governed by spatial scales not resolved in regional climate models and adjustments have to be made to overcome this (e.g., Fiddes and Gruber, 2014). A variety of methods has been developed for this purpose, differing in terms of data requirements and computational cost. While empirical-statistical scaling requires reference data for training and assumes

(c) Author(s) 2019. CC BY 4.0 License.





a temporal robustness of the employed statistical relations (e.g., Ehret et al., 2012; Maraun, 2013), dynamic downscaling by means of high-resolution atmospheric modelling has high computational costs (e.g., Gutmann et al., 2016).

In this paper, we present Sval_Imp_v1 (Schuler, 2018), a high resolution gridded dataset obtained using a hybrid methodology combining multi-dimensional interpolation with simple dynamical modelling. The dataset is defined on a 1km grid
covering the archipelago of Svalbard, located in the Norwegian Arctic (74-82°N, 10-35°E). The atmospheric reanalyses ERA-40 and ERA-interim by the European Centre for Medium-Range Weather Forecasting have been downscaled to cover the period 1957-2017 at a resolution of 6h. The dataset comprises the near-surface variables required to compute the surface energy balance, namely air temperature, precipitation, relative humidity, wind speed, and downwelling components of shortwave (solar) and longwave (thermal) radiation. Sval_Imp_v1 is publicly available from the Norwegian Research Data Archive NIRD, a data
repository (https://doi.org/10.11582/2018.00006). In the following, we describe the methodology used to derive the dataset, present the organization of the data in the repository and discuss the performance of the downscaling procedure. For the latter, the dataset is compared to a wealth of data available from long-term operational as well as short-term scientific records of meteorological and glaciological measurements. Whereas operational records are biased to low-elevations around the fringes of the archipelago, we stress the hitherto under-used potential of project-based measurements in the interior, high-elevation
regions for evaluating atmospheric models. For instance, records of snow accumulation on large ice masses may represent measures of seasonally-integrated precipitation in regions sensitive to the downscaling procedure, thus providing added-value. Sval_Imp_v1 has been employed entirely or in parts by a range of projects for forcing process models of the surface energy and mass balances of glaciers (Østby et al., 2017), precipitation patterns and meltwater production in the Kongsfjord area (Pramanik et al., 2018), for assimilation of remotely-sensed snow cover using a snow distribution model (Aalstad et al., 2018) as
well as for assessing growing conditions for fungi (Botnen et al., in prep.). Further, the dataset has been used to assess changes and trends in climate conditions of Svalbard (Hanssen-Bauer et al., 2019).

## 2 Methodology

To generate fields of near-surface air temperature, precipitation, relative humidity, wind and downwelling shortwave and longwave radiation, we have downscaled the ERA-40 and ERA-interim reanalyses of the European Centre for Medium-Range
Weather Forecasts (Uppala et al., 2005; Dee et al., 2011). The reanalysis data is provided at 6-hour intervals and has been retrieved on a $0.75° \times 0.75°$ spatial grid (Figure 1), covering the periods 1957-2002 (ERA-40) and 1979-2017 (ERA-interim).

### 2.1 Downscaling

Precipitation is often heavily biased in coarsely-resolved reanalyses, especially in environments with pronounced topography, where it typically is too low and lacks spatial detail (Schuler et al., 2008). This is associated with the smoothed representation
of the actual topography in the large-scale model used for the reanalysis (Figure 1), leading to an underestimate of orographic precipitation. Figure 1 shows that ERA greatly generalizes the high resolution topography, representing Svalbard as a wide and flat bump, that exceeds sea-level far off the actual coastlines, while surface elevation in the interior does not exceed 400 m asl.

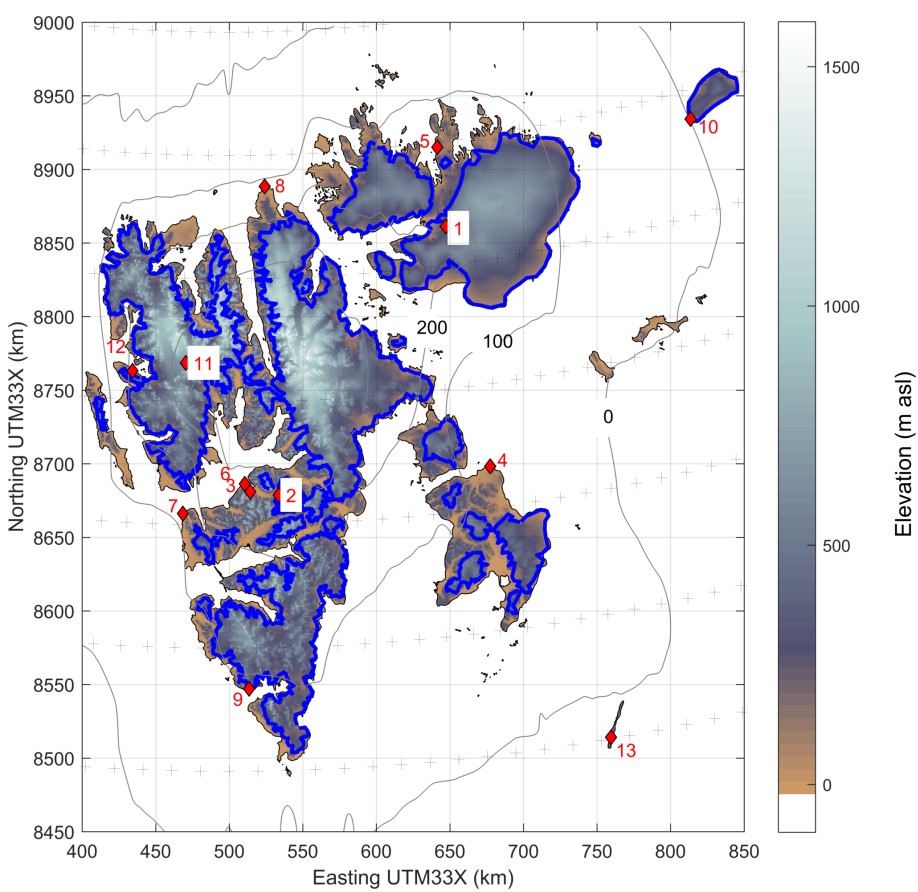

**Figure 1.** The map shows the Svalbard archipelago, shading indicates surface elevation, glacierized areas are outlined in blue. The locations of meteorological stations are represented by red diamonds, the numbers refer to the station names: 1 - Etonbreen, 2 - Janssonhaugen, 3 - Gruvefjellet, 4 - Kapp Heuglin, 5 - Rijpfjorden, 6 - Svalbard Airport, 7 - Isfjord Radio, 8 - Verlegenhuken, 9 - Hornsund, 10 - Kvitøya, 11 - Holtedahlfonna, 12 - Ny-Ålesund, 13 - Hopen. The black crosses indicate the grid points of the ERA reanalyses, and the grey countourlines indicate the topography of Svalbard in the ERA reanalyses at 0, 100, 200 and 300 m asl.





In contrast, elevation in our 1×1 km topography peaks 1600 m asl. The roughness of the actual topography that gave rise to the name of the main island 'Spitsbergen' (sharp tops) is not present in the smoothed topography used for the ERA reanalyses. We assume that this is the main reason for the poor performance of reanalyzed precipitation, and use instead a linear theory (LT) of orographic precipitation (Smith and Barstad, 2004) to account for orographic enhancement.

The other required climate variables are downscaled to the 1 km grid largely following the TopoSCALE methodology (Fiddes and Gruber, 2014) which also builds on the assumption that weaknesses in the representation of topography at the coarse scale are mainly responsible for the misfit between coarse scale and point observations. TopoSCALE exploits the relatively high vertical resolution of the reanalysis data to downscale variables to the elevation of the actual topography, based on the properties of the vertical structure in the reanalysis. The downscaled fields preserve the horizontal gradients present in

ERA, but include additional features caused by the real topography not present in ERA (Figure 1). In doing so, we add spatial detail to the reanalysis fields that is consistent with the temporal evolution of atmospheric conditions of the reanalysis. This approach is assumed to outperform simpler bias corrections, since transient properties of the atmosphere are accounted for. For example, transient lapse rates including inversions in the reanalysis data will be preserved in the downscaled product.

In our application, we modified the TopoSCALE methodology regarding downscaling of direct shortwave radiation and air

temperature, as described in the following.

## 2.2   Precipitation

The LT-model describes an air parcel as it moves across a prescribed surface topography. The air parcel is characterized by its temperature, stability, wind direction and speed. Terrain induced uplift of the air parcel results in condensation and eventually precipitation of moisture downstream of the uplift. This model has been successfully evaluated using precipitation gauges

(Barstad and Smith, 2005) and snow measurements (Schuler et al., 2008; Østby et al., 2017) and applied for downscaling precipitation (e.g. Crochet et al., 2007; Jarosch et al., 2012; Roth et al., 2018). The linear theory utilizes a Boussinesq description of mountain wave to derive a transfer function that, for given wind conditions, relates the orographically enhanced precipitation to terrain topography.

By spectral decomposition and algebraic manipulation, Smith and Barstad (2004) derived the following transfer function

$$\hat{P}(k,l) = \frac{C_w i \sigma \hat{h}(k,l)}{(1 - imH_w)(1 + i\sigma\tau_f)(1 + i\sigma\tau_c)}. \tag{1}$$

relating $\hat{P}(k,l)$, the Fourier-transform of the precipitation enhancement, to the Fourier-transform of terrain elevation $\hat{h}(k,l)$, with $k$ and $l$ being the horizontal wave numbers. This relation depends on the uplift sensitivity factor $C_w$, thickness of the moist layer $H_w$, the intrinsic frequency $\sigma(k,l) = Uk + Vl$ ($U$ and $V$ being the east and north components of the wind vector), and the conversion and fallout time scales $\tau_c$ and $\tau_f$, respectively. In Equation (1), the vertical wave number $m$ controls the depth

and tilt of the forced air uplift and is a function of the moist Brunt-Väisälä frequency, $N_m$, a quantity describing atmospheric stability.

Precipitation rates are obtained by retransforming $\hat{P}$ and adding it to the background precipitation $P_\infty$, that accounts for large-scale frontal and convective precipitation separate from orographic precipitation $P_{\text{oro}}$. $P_\infty$ has been corrected for the





orographic effect already present in the ERA reanalyses $P_{\mathrm{oro}}(h_{\mathrm{ERA}})$ by estimating this effect applying Equation 1 to the large-scale topography $h_{\mathrm{ERA}}$ and removing the result from the ERA precipitation $P_{\infty} = P_{\mathrm{ERA}} - P_{\mathrm{oro}}(h_{\mathrm{ERA}})$ (Schuler et al., 2008). Total precipitation, $P_{\mathrm{total}}$, is then

$$P_{\mathrm{total}}(x,y) = \max\left[f \iint \hat{P}(k,l)e^{i(kx+yl)}dkdl + P_{\infty}, 0\right]. \tag{2}$$

Since the theory assumes saturated conditions, we account for reduced orographic enhancement at lower humidity by adopting a correction factor $f$ proposed by Sinclair (1994)

$$f = \begin{cases} \left(\frac{RH-0.8}{0.2}\right)^{1/4} & : \quad RH \geq 0.8 \\ 0 & : \quad \text{otherwise} \end{cases} \tag{3}$$

which suppresses orographic enhancement when $RH < 0.8$.

Instead of treating $N_m$, $\tau_f$ and $\tau_c$ as adjustable, constant parameters, we exploit the evolution of the moisture bearing layer of the atmosphere, described in the reanalyses to derive transient values. In doing so, we remove calibration parameters from our method and enable weather dependent variation of $N_m$, $\tau_f$ and $\tau_c$. Values of $N_m$ are calculated following

$$N_m^2 = \frac{g}{\overline{T}}(\Gamma_m - \Gamma_e) \tag{4}$$

where $g$ is gravitational acceleration and $\overline{T}$ is vertically averaged air temperature weighted by the moisture content at several pressure levels (Jarosch et al., 2012). Environmental lapse-rates $\Gamma_e$ are derived from air temperature at 700 hPa and 850 hPa and corresponding geopotential heights. $\Gamma_m$ is the moist adiabatic lapse rate, calculated according to Stone and Carlson (1979) using vertically averaged values of atmospheric properties from the reanalyses weighted by moisture content. This follows the convention that a positive lapse rate represents cooling with increasing elevation. Barstad and Smith (2005) report that typical values of $N_m$ range between 0 s$^{-1}$, representing an atmosphere with no stratification, and 0.01 s$^{-1}$ representing a stably stratified atmosphere. To avoid conditions inconsistent with the assumptions of the theory, we limit $N_m$ to this range . The quantities $H_w$, $C_w$ and $m$ are derived from $N_m$ (Smith and Barstad, 2004).

Advection time scales $\tau_c$ and $\tau_f$ are assumed equal and $\tau = \tau_c = \tau_f = H_w/v$ is derived from the thickness of the moist layer $H_w$ and accounting for a typical hydrometeor fall speed $v$, which is taken as constant but allowed to take different values for solid and liquid hydrometeors. This phase transition is determined by a threshold temperature of 273 K, hence $v(T \leq 273K) = 1\,\mathrm{m\,s^{-1}}$ for solid and $v(T >= 273K) = 2\,\mathrm{m\,s^{-1}}$ for liquid precipitation. In Equation 1, terrain elevation $h$ is the only gridded variable, the other variables represent averages over the volume of the described air parcel. To characterize this air parcel, we first vertically average the values defined at the nodes of the horizontal domain over the 700 hPa and 850 hPa pressure-levels weighted by the moisture content of the individual layers. These vertically averaged values are then horizontally averaged over an area defined by a 200 km buffer around the 200 m contour of the reanalysis topography, the latter roughly outlining the extend of the archipelago (Fig. 1).

This setup is then applied to each 6h timestep and the resulting timeslices are progressively added to the record.



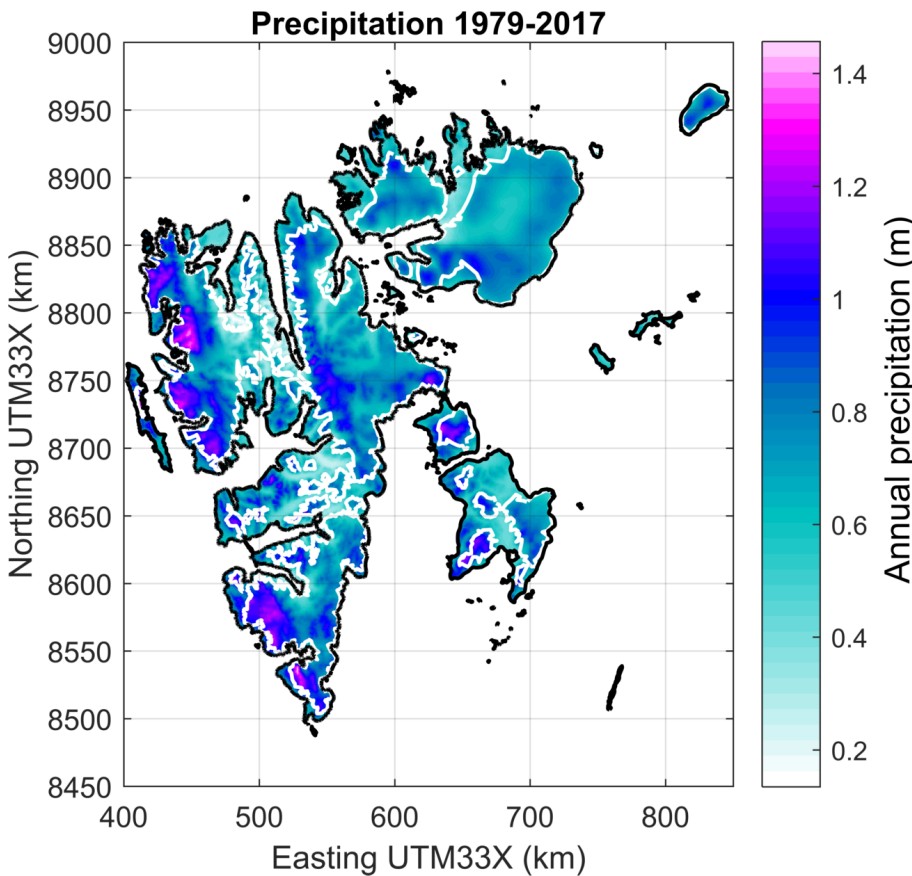

**Figure 2.** Example plot of longterm mean annual precipitation (m) over 1979-2017, downscaled from ERA-interim.

## 2.3 Temperature

The downscaling for near-surface air temperature at 2 m level ($T2$) closely follows the TopoSCALE procedure Fiddes and Gruber (2014), thereby we assume that the vertical structure of the free atmosphere determines the distribution of $T2$ with terrain elevation. For each 6h-time step, $T2$ is derived from a three-dimansional interpolation of the vertical temperature structure of the large-scale reanalysis to the location of the grid nodes representing the high-resolution terrain elevation. We notice, that for a melting snow or ice surface, skin temperature is bounded to 273 K, influencing the near surface air temperature and resulting in reduced along-surface temperature lapse-rates (e.g., Marshall et al., 2007). To account for this effect, we apply a simple horizontal interpolation of $T2_{\mathrm{ERA}}$ where $T2 > 273$ K. This strategy is motivated by the discovery of unrealistic warm temperatures at higher elevations in a test application. The occurrence of this unphysical temperature inversion was restricted to the melting period, caused by extrapolation of surface inversions close to a snow/ ice surface the temperature of which is capped at 273 K. To avoid this effect, we assume that where $T2 > 273$ K, the $T2$ of the reanalysis is consistent with a melting surface and hence more realistic than a free-atmosphere interpolation.


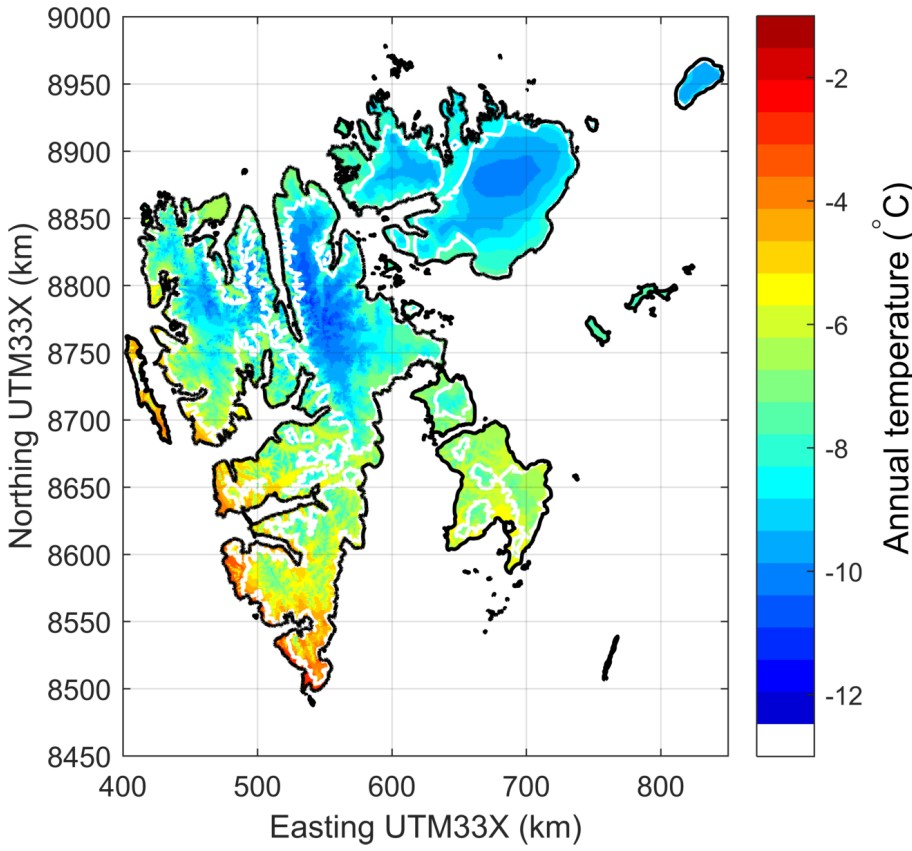

**Figure 3.** Example plot of mean air temperature (2 m) in $^{\circ}C$, over the period 1979-2017, downscaled from ERA-interim.

## 2.4 Radiation

Downscaling of shortwave and longwave downwelling radiative fluxes ($SW$ and $LW$, respectively) was conducted by adopting the TopoSCALE methodology (Fiddes and Gruber, 2014) with a few adjustments. The shortwave radiation at surface level of the reanalysis is projected to the high-resolution topography in a three-step procedure: first the surface $SW$ flux is separated into direct and diffuse components; second, the direct component is corrected for the elevation difference between the reanalysis surface and the high-resolution topography considering an effective atmospheric transmissivity that is derived from top-of-atmosphere and surface fluxes; third a topographic correction is applied to account for effects of slope and aspect of the high-resolution topography, as well as shading by surrounding topography. To compute direct solar radiation we apply the relationship of Kumar et al. (1997) for atmospheric attenuation rather than the one given by Fiddes and Gruber (2014). Solar geometry variables such as solar zenith and azimuth, and topographic shading due to local slope and aspect are calculated following Reda and Andreas (2004). Cast shadow and hemispherical obstructions caused by surrounding topography are calculated following Ratti (2001). The longwave surface flux is downscaled by correcting for the elevation difference between reanalysis and high-resolution grids using an atmospheric emmissivity. This emissivity is estimated by accounting for a clear-

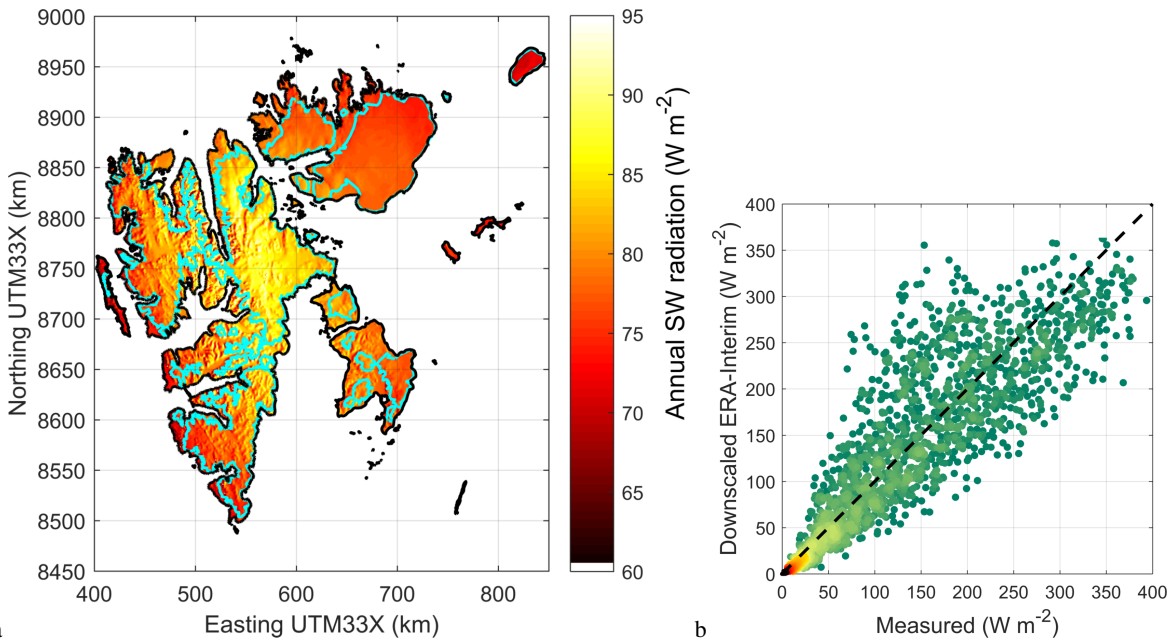

**Figure 4.** a) Example plot of 1979-2017 incident shortwave radiation ($\mathrm{W\,m^{-2}}$), downscaled from ERA-interim; b) comparison between measured (Maturilli et al., 2013) and downscaled daily values of SW at Ny-Ålesund. The color indicates the density of points from red (high) to green (low).

sky component that depends on humidity and temperature and a cloud component that is estimated from the difference between the clear-sky component and the reanalysis longwave flux. Further terrain effects are incorporated through multiplication with the sky-view factor.

## 2.5 Relative humidity and windspeed

5 Similar to our assumption about $T2$ over a melting surface, we suggest that windspeed and $RH$ in the boundary layer are more affected by surface rather than by free-atmosphere conditions and we hence apply a simple 2D interpolation of the near-surface values of the reanalysis.

## 3 Performance evaluation

Østby et al. (2017) and Vikhamar-Schuler et al. (2019) have conducted thorough evaluation of the Sval_Imp_v1 dataset using
10 data from meteorological stations. For details, we refer to the paper by Østby et al. (2017); in this paper, we summarize their main results and present additional evaluation of the precipitation using snow measurements.





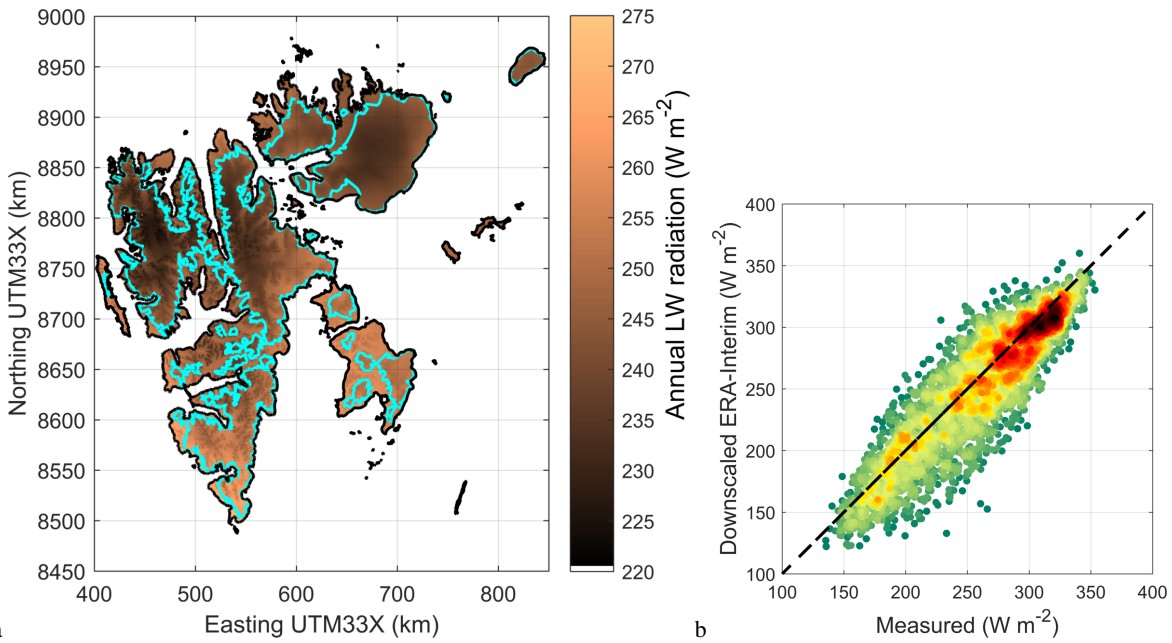

**Figure 5.** a) Example plot of 1979-2017 downwelling longwave radiation ($W \, m^{-2}$), downscaled from ERA-interim; b) comparison between measured (Maturilli et al., 2013) and downscaled daily values of LW at Ny-Ålesund. The color indicates the density of points from red (high) to green (low).

## 3.1 Precipitation

At the operational weather stations (Fig. 1), downscaled precipitation (Fig. 2) is overestimated by 5 to 25 mm per month at the weather stations, with a slightly higher bias during winter (Table 1). This is also consistent with the findings of Vikhamar-Schuler et al. (2019) who evaluated the performance at 6 weather stations for several 30-year reference periods (1961-1990, 1971-2000, and 1988-2017). These biases are partly caused by too low precipitation measurements, the values of which are heavily affected by wind-induced undercatch, especially for solid precipitation. Førland and Hanssen-Bauer (2000) suggest that for solid precipitation, the actual precipitation at wind-exposed sites may be up to 80% higher than the gauge record. A newly developed correction scheme for Norwegian mountain environments (Wolff et al., 2015) supports the finding of large undercatch for solid precipitation, however, this corrections has not yet been applied for arctic conditions in Svalbard.

In addition to gauge measurements from low-elevation stations along the coast, we also used snow survey transects across Austfonna, a large ice cap in NE Svalbard (Fig. 1) to evaluate the Sval-Imp precipitation. We suggest that snow deposition on large glacier areas, measured at the end of the winter, represents seasonally integrated precipitation. While these measurements do not allow temporal resolution below one snow season (typically Oct-May), they provide useful information about spatial precipitation patterns, far off the operational meteorological stations. There is generally good agreement concerning the spatial pattern across the Austfonna ice cap (Fig. 6), where snow accumulation reveals a distinctive SE-NW asymmetry (Taurisano

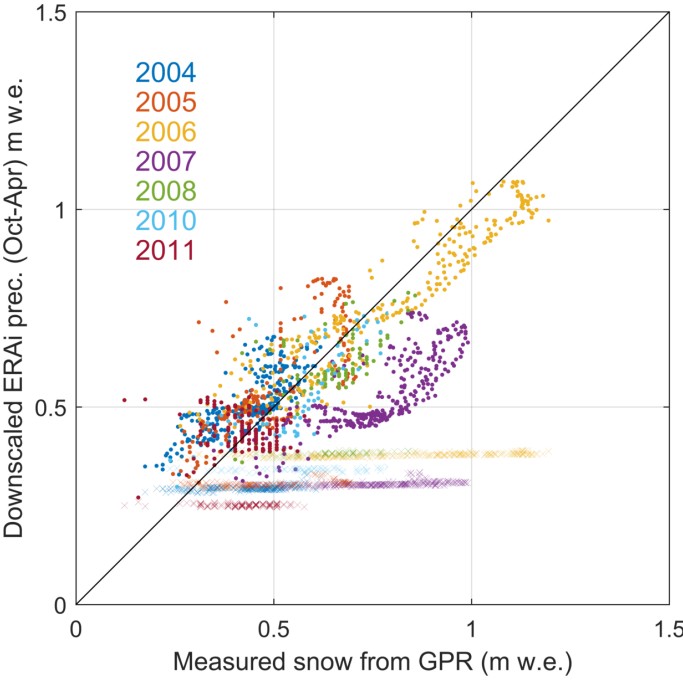

**Figure 6.** Comparison of snow water equivalent from gound-penetrating radar (GPR) transects across the Austfonna ice cap (Taurisano et al., 2007; Dunse et al., 2009) with the seasonally (Oct - Apr) accumulated precipitation product. Dots refer to the downscaled winter precipitation, whereas crosses represent unscaled ERA-interim winter precipitation.

et al., 2007; Schuler et al., 2007; Dunse et al., 2009) caused by orographic enhancement of precipitation coming from the SE sector (i.e., the Barents Sea), although in individual years the downscaled values underestimate the measured accumulation, especially in 2007 (Fig. 6). Even though, there is considerable scatter between observed and downscaled winter precipitation, there is positive correlation, indicating that the spatial pattern is matched and in most years there is no systematic bias,
5   showing that the overall precipitation amount is adequately represented. On the other hand, the unscaled ERA-interim winter precipitation shows almost no spatial variation and considerably underestimates observed values (Fig. 2).

Østby et al. (2017) found that the winter mass balance of Hansbreen was not well reproduced both in terms of spatial pattern as well as accumulation amount. Aas et al. (2016) similarly reported the lowest performance for Hansbreen although they had used a much more complex precipitation scheme than the one presented here. The generally low performance of several
10   precipitation distribution schemes compared to the Hansbreen record has been interpreted to result from local conditions at Hansbreen where the spatial distribution of snow is caused by wind redistribution rather than by the spatial precipitation pattern Grabiec et al. (2006).





## 3.2 Temperature

Downscaled temperatures (Fig. 3) are compared to observations at the meteorological stations listed in Table 1 mostly for the period after 2004. Despite altitude differences of up to 100 m between measuring site and corresponding grid node in the model, no altitude correction is performed, due to unknown lapse rates. In general the agreement is good between downscaled ERA and

observed air temperatures, with biases mostly below 1.5 K (Tab. 1). Despite a small bias for mean annual temperatures, there is a clear seasonal bias, with ERA temperatures too warm during winter and too cold during summer (Table 1). Although the biases are negative during summer, ERA is too warm over the glaciers during summer, when 2-m air temperatures are above freezing. These findings are consistent with those of Vikhamar-Schuler et al. (2019) who evaluated differences in seasonal mean values for different 30-year periods 1961-1990, 1971-2000 and 1988-2017.

At Svalbard Airport, the performances of downscaled ERA-40 and ERA-interim are investigated for the entire model period. Over 1957-1979 only monthly measured temperatures are available at Svalbard Airport, where downscaled ERA-40 has a monthly root mean square error (RMSE) of 2.3 °C. For the 1979-2002 period the reanalysis products overlap with monthly RMSE of 1.8 °C and 1.5 °C at Svalbard Airport for ERA-40 and ERA-interim, respectively. We attribute the lower performance prior to 1979 to the lack of satellite observations to constrain sea surface temperatures and sea ice cover in the reanalysis. Since

the Svalbard Airport temperature record and other sites at the west coast are likely incorporated into the reanalysis, the quality of the reanalysis in the pre-satellite era is possibly even lower in remote areas with no observations. The annual observed air temperature trend for the period 1957-2013 at Svalbard Airport is $0.70\pm0.22$ °C decade$^{-1}$, while the downscaled ERA data has an insignificantly lower warming trend of $0.67\pm0.19$ °C decade$^{-1}$ at Svalbard Airport.

## 3.3 Radiation

To evaluate the quality of Sval-Imp radiation components, we use available records from two stations, roughly 300 km apart from each other. The record from Ny-Ålesund is from a daily serviced Baseline Surface Radiation Network station (Maturilli et al., 2013) whereas the Austfonna measurements are collected by an autonomously recording weather station (Schuler et al., 2014).

In Ny-Ålesund the model largely reproduces observations both for short and longwave radiation (Figs. 5 and 4, Tab. 1).

During winter, downwelling longwave radiation is slightly underestimated, while there is no bias during summer. Since there is no temperature bias in Ny-Ålesund during winter, the underestimation of longwave radiation is indicative of a too thin cloud cover in the reanalysis. In general, the representation of clouds are among the major issues of the reanalysis (Aas et al., 2016). Downwelling shortwave radiation is overestimated by 7 W m$^{-2}$ over the summer season in Ny-Ålesund. There is a much better agreement with radiation observations in Ny-Ålesund than on Etonbreen (Fig. 1), in northeastern Svalbard. This is to

be expected, since radio soundings and other observation data from Ny-Ålesund are assimilated into ERA-interim. Therefore, cloud cover at Ny-Ålesund is much better represented by the reanalysis than at Austfonna. On Etonbreen during summer, Sval-Imp underestimates downwelling shortwave radiation by 40 W m$^{-2}$ while downwelling longwave radiation is overestimated by 12 W m$^{-2}$, both indicative of a too thick atmosphere or too many clouds in the reanalysis. However, these biases could also





be partly explained by measurement uncertainty caused by rime on the sensor or by sensor tilt. The latter issue is caused by the fact that the ice foundation of an autonomous weather stations may melt and deform causing tilt and thereby large errors, especially at high solar zenith angles (Bogren et al., 2016).

### 3.4 Relative humidity and windspeed

5 For relative humidity the reanalysis represents the seasonality well, and in late summer both the humidity and the biases are of largest magnitude. At the two coastal stations at Hopen and Rijpfjorden, the downscaled reanalysis is too dry whereas it is too humid at the two higher elevation stations. The coarse land mask of the reanalysis and the poor representation of sea ice are most likely the main causes for these biases.

Wind speeds are reproduced reasonably well, including the seasonal cycle (Tab. 1). Biases are within $\pm 1.5 \ \mathrm{m \, s^{-1}}$ with no 10 clear seasonal trend. It is likely that the biases are caused by site specific effects, such as deceleration of air flow in the lee of a topographic obstacle or acceleration due to channelizing through valleys.

### 4 Dataset structure

The downloadable dataset comprises individual files for each of the variables precipitation, temperature, relative humidity, windspeed, incident shortwave and downwelling longwave radiation. The records are organized in one file per month for each 15 of the reanalysis periods; ERA-40: September 1957 to August 2002, and ERA-interim: January 1979 to December 2017. Each file contains the discovery metadata, and the complete metadata to locate the stack of fields in space and time. The grid is regular and rectangular in UTM33X projection, the coordinates of which are defined by the vectors X (m easting in UTM33X, 448 elements) and Y (m northing in UTM33X, 548 elements). In geographical coordinates the grid is non-regular and therefore the location of each grid node is defined, rendering Latitude and Longitude (in decimal degrees) each a 448×548 20 matrix. The timestamp is given in days since 1 January 1900 using a standard Gregorian calendar having 365 days per year, i.e. without accounting for leap years. In addition, there is one file containing the stationary fields, i.e. the surface topography and land-ocean mask. The file format is netCDF according to the CF conventions (http://cfconventions.org/), with all required metadata included. The metadata adhere to ISO19115 geospatial metadata standards and the Directory Interchange Format (DIF) requirements of the Global Change Master Directory GCMD (https://gcmd.nasa.gov/DocumentBuilder/defaultDif10/ 25 guide/index.html), and global attributes comply to the Attribute Convention for Data Discovery ACDD (http://wiki.esipfed. org/index.php/Attribute_Convention_for_Data_Discovery_1-3).

Table 2 gives an overview over the number of files and their sizes for the different epochs (ERA-40, ERA-interim) and variables contained in the dataset. The naming convention for the individual files is <EPOCH>_<VAR>_<YYYYMM>.nc, where EPOCH is either "ERA40" or "ERAi", VAR is an abbreviation of the variable of interest (one of "precip", "temp", "RH", 30 "windspeed", "SWi" or "LWi") and YYYYMM identifies year and month, for instance *ERAi_temp_200410.nc* is the name of the file containing temperature from ERA-interim for October 2004.





**Table 1.** Meteorological stations used for validation of the downscaled reanalysis and their elevation (second line). $N$ indicates the number of daily averages used in the validation. Seasonal biases in meteorological variables (downscaled minus observational averages) at all sites are averaged over each site's observation period. Shown are air temperature T (K), relative humidity, RH (%), wind speed, WS (ms$^{-1}$), shortwave radiation, SW, and longwave radiation, LW (both in Wm$^{-2}$) and precipitation, P (mm). Column headings $S$ and $W$ denote summer (Jun-Aug) and winter (Sep-May), respectively. Positive numbers indicate that the model results are larger than the observations. The second row at each site is the bias between the raw ERA data and the observations.

| Location | ΔT | | | ΔRH | | | ΔWS | | | ΔSW | | | ΔLW | | ΔP | | |
| --- | --- | --- | --- | --- | --- | --- | --- | --- | --- | --- | --- | --- | --- | --- | --- | --- | --- |
| | S | W | $N_T$ | S | W | $N_{RH}$ | S | W | $N_{WS}$ | S | W | $N_{rad}$ | S | W | S | W | $N_P$ |
| Etonbreen[†] | 0.2 | 1.3 | 3295 | -2.2 | -5.4 | 2738 | 0.3 | 0.2 | 2913 | -40 | -10 | 3240 | 12 | -14 | – | – | 0 |
| 369 m asl | *1.1* | *1.9* | | *-2.2* | *-5.4* | | *0.3* | *0.2* | | *-37* | *-9* | | *17* | *-16* | – | – | |
| Janssonhaugen | -1.1 | 0.3 | 910 | – | – | 0 | -1.8 | -1.3 | 945 | – | – | 0 | – | – | – | – | 0 |
| 270 m asl | *-1.0* | *-0.6* | | – | – | | *-1.8* | *-1.3* | | – | – | | – | – | – | – | |
| Gruvefjellet | 0.2 | 0.8 | 2555 | 3.1 | -2.9 | 2555 | 0.0 | -0.2 | 2551 | – | – | 0 | – | – | – | – | 0 |
| 464 m asl | *0.7* | *0.9* | | *3.1* | *-2.9* | | *0.0* | *-0.2* | | – | – | | – | – | – | – | |
| Kapp Heuglin | -0.0 | 0.7 | 2099 | – | – | 0 | -0.0 | 1.0 | 2112 | – | – | 0 | – | – | – | – | 0 |
| 18 m asl | *0.3* | *1.0* | | – | – | | *-0.0* | *1.0* | | – | – | | – | – | – | – | |
| Rijpfjorden | -0.3 | 0.9 | 1495 | 5.7 | 2.6 | 1495 | 0.8 | 0.9 | 1304 | – | – | 0 | – | – | – | – | 0 |
| 10 m asl | *0.0* | *0.6* | | *5.7* | *2.6* | | *0.8* | *0.9* | | – | – | | – | – | – | – | |
| Svalbard Airport | -2.4 | -0.4 | 12777 | – | – | 0 | -1.3 | -1.0 | 12724 | – | – | 0 | – | – | 25 | 26 | 199* |
| 28 m asl | *-2.4* | *-2.2* | | – | – | | *-1.3* | *-1.0* | | – | – | | – | – | – | – | |
| Isfjord Radio | -1.6 | -1.2 | 1666 | – | – | 0 | – | – | 0 | – | – | 0 | – | – | – | – | 0 |
| 13 m asl | *-1.5* | *-0.5* | | – | – | | – | – | | – | – | | – | – | – | – | |
| Verlegenhuken | -0.5 | 0.0 | 986 | – | – | 0 | -1.9 | -1.7 | 1700 | – | – | 0 | – | – | – | – | 0 |
| 8 m asl | *-0.3* | *0.5* | | – | – | | *-1.9* | *-1.7* | | – | – | | – | – | – | – | |
| Hornsund | -0.2 | 0.4 | 4635 | – | – | 0 | -0.0 | 0.4 | 4473 | – | – | 0 | – | – | 5 | 19 | 95* |
| 10 m asl | *-0.0* | *0.8* | | – | – | | *-0.0* | *0.4* | | – | – | | – | – | – | – | |
| Kvitøya | -0.1 | -0.1 | 740 | – | – | 0 | -0.9 | -1.3 | 702 | – | – | 0 | – | – | – | – | 0 |
| 10 m asl | *0.3* | *0.6* | | – | – | | *-0.9* | *-1.3* | | – | – | | – | – | – | – | |
| Holtedahlfonna[†] | 1.3 | – | 317 | – | – | 0 | -0.7 | – | 265 | – | – | 0 | – | – | – | – | 0 |
| 688 m asl | *3.2* | – | | – | – | | *-0.7* | – | | – | – | | – | – | – | – | |
| Ny-Ålesund | -1.7 | -0.0 | 12666 | 7.6 | 3.2 | 12708 | 0.7 | 0.8 | 12349 | -7 | 4 | 3652 | 2 | 9 | 23 | 18 | 212* |
| 8 m asl | *-1.6* | *-1.2* | | *7.6* | *3.2* | | *0.7* | *0.8* | | *-8* | *3* | | *0* | *15* | – | – | |
| Hopen | 1.9 | 5.2 | 13178 | – | – | 0 | 0.3 | -0.1 | 13044 | – | – | 0 | – | – | 7 | 10 | 306* |

*: Number of months

[†]:Station located on glacier.



**Table 2.** Overview file structure of the Sval_Imp_v1 dataset

| Sval_Imp_v1 | | | | | | | | | | | | |
|---|---|---|---|---|---|---|---|---|---|---|---|---|
| Stationary fields | Svalbard_DEM_mask_pcorr.nc | | | | | | | | | | | |
| Variables | T2 | | P | | WS | | RH | | SW | | LW | |
| Reanalysis | ERA40 | ERAint | ERA40 | ERAint | ERA40 | ERAint | ERA40 | ERAint | ERA40 | ERAint | ERA40 | ERAint |
| Number of files | 540 | 468 | 540 | 468 | 540 | 468 | 540 | 468 | 540 | 468 | 540 | 468 |
| Size (GB) | 22.8 | 19.8 | 33.1 | 28.8 | 34.1 | 29.8 | 25.6 | 22.6 | 21.7 | 19.0 | 31.3 | 27.5 |

## 5 Conclusions

We present a gridded dataset of near-surface, meteorological variables at 1km resolution covering the Svalbard archipelago. The set of variables enables application of energy balance models and comes at a time steps of 6 h. The high-resolution grids are derived from coarse-scale reanalyses ERA-40 for the period 1957-2002 and from ERA-interim for 1979-2017. We describe the intermediate-complexity downscaling procedure used to generate this dataset. Furthermore, we evaluate the performance of the downscaled data using a suite of different meteorological and glaciological measurements and refer to several applications of this dataset in different disciplines, all of them requiring longterm coverage at small spatial scales.

## 6 Data availability

The dataset is openly available from the National e-Infrastructure for Research Data (NIRD) archive at https://doi.org/10.11582/2018.00006 and referred to as *Svalbard impact assessment forcing dataset, version 1*, (Schuler, 2018).

ERA-40 (Uppala et al., 2005) and ERA-interim (Dee et al., 2011) data were retrieved from the ECMWF Public Datasets web interface at https://apps.ecmwf.int/datasets/.

Weather station data are provided by the Norwegian Meteorological Institute and are available at https://eklima.met.no, and by the University Centre of Svalbard at https://www.unis.no/resources/weather-stations/. Radiation from the BRSN-station in Ny-Ålesund are provided by Maturilli et al. (2014).

*Author contributions.* TVS retrieved reanalysis data, developed code and performed the downscaling for precipitation, temperature, RH and windspeed; TIØ developed code and performed downscaling of radiation components. Evaluation performance was conducted collaboratively. TVS was responsible for formatting, providing metadata and archiving the dataset; he led writing of the manuscript, TIØ contributed to writing.

*Competing interests.* There are no competing interests present.



*Acknowledgements.* We greatly acknowledge the European Centre for Medium-range Weather Forecast, ECMWF for access to their reanalysis products ERA-40 and ERA-interim. The Norwegian Meteorological Institute has been pioneering and is exemplary for open access to historical data, especially through their eklima-service. Field work on Austfonna has been supported by NFR through IPY-Glaciodyn, EU-ice2sea and ESA-CRYOVEX, along with KD scholarship for TIØ and mobility funds from SVALI.



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
