# Peer review of "Sval\_Imp: A gridded forcing dataset for climate change impact research on Svalbard"

_Earth System Science Data, 2019_

## Referee Comment (RC1) · Kabir Rasouli (Referee) · 11 Dec 2019

Summary:

In this manuscript, downscaled meteorological variables with 1km horizontal and 6hr temporal resolutions were presented for Svalbard in the Norwegian Arctic. The gridded product was obtained from downscaling reanalysis products of ERA40 and ERA-Interim using mountain wave linear theory and validating against observations recorded at the meteorological stations. The method and data used are robust and the dataset presented is very useful for testing the models and diagnosing the changes in the Arctic. The manuscript has the potential to be published after a minor revision. Below are

some comments that can be used to improve the presentation of the manuscript.

General comments:

1. The accumulated snow on ice cannot be representative of precipitation as snow cover and depth on ice are not uniform because of blowing wind effect on redistribution of snow, roughness of ice surface, and spatial variability of blowing snow sublimation (Line 11-12 and Fig. 6). This is clearly shown by a poor relationship between unscaled ERA-interim winter precipitation and snow water equivalent on Figure 6. Either clarify this or remove the text and figures related to this argument.

2. It is not clear if monthly data for precipitation were used for downscaling the ERA reanalysis products or daily? If monthly data were used, how reliable is monthly-to-6hourly temporal downscaling? If daily data were used, why number of months for precipitation is provided in Table 2? For other variables if daily variables recorded at the stations were used how were the daily values disintegrated to 6-hourly data?

3. I am wondering how many of the stations that were used for validation of the down-scaled meteorological variables were originally used to assimilate the reanalysis products. Is it only the Ny-Ålesund station? If the station data have been used in assimilating reanalysis products, it is not surprising to see high correlations between time series of ERA interim grids and stations.

Specific comments: (P: Page, L: Line)

1. I recommend removing the index of "Sval_Imp_v1" form the title and across the manuscript.

2. P 2 L 13: Rewrite "Whereas operational records are biased to low-elevations . . .".

3. P 4 L 1: Replace "In contrast, elevation in our 1×1 km topography peaks 1600 m asl" with "In contrast, the highest elevation in our gridded topography map is 1600 m asl".
4. P 4 L 2: Check the grammar in "present" and also in the next line in "and use". Do you mean you have used linear theory?

5. P4 L10: Add ", which was" before "not present in the ERA products".

6. P4, L14-15: The authors have mentioned "we modified the TopoSCALE methodology regarding downscaling of direct shortwave radiation and air temperature, as described in the following." However, the following section is about precipitation and not about radiation or temperature. Use more specific section numbers or names.

7. Use "air" temperature instead of temperature when you refer to air temperature. This can be mixed up with surface or soil temperatures if not specified.

8. P10 L12: Change "Grabiec et al. (2006)." to "(Grabiec et al., 2006)."

9. Is the dataset presented in this paper and stored in the website only downscaled products or both raw unscaled and processed downscaled data (e.g., Table 2)?

10. Check for the typos and English grammar errors across the manuscript.

―――――――――――――――――――――――

---

## Referee Comment (RC2) · Andrew Newman (Referee) · 30 Dec 2019

General comments:

This paper presents a 1 km gridded meteorological dataset for Svalbard that is generated through a hybrid statistical-simple dynamical method. They downscale ERA-40 and ERA-Interim data from 1957-2017 using linear mountain wave theory for precipitation and a combination of simple 2-D and more complex 3-D interpolation for other meteorological variables such as temperature and radiation. The base methods used for this dataset have been published previously, there are only minor modifications here.

The performance evaluation given by the authors builds on previous studies and uses snow surveys to evaluate precipitation in more detail. I find this particularly useful and a good addition to the dataset documentation. More clarification of methods and the product evaluation should be included so users have a more complete reference paper.

Specific comments:

1) Although the authors give citations to other papers that evaluate this product, it would be beneficial to have a more explicit summary of those results in this paper so they are more readily available in one location.

2) The paper needs more discussion of the differences in performance between the ERA-40 and ERA-Interim time periods using the overlap period wherever possible. This will be very useful to users of the full dataset. There is some discussion of this for temperature, but not for any other variables.

3) Are there any other snow transect data available that could be added to this analysis? It appears that another paper evaluated this product on another glacier named Hansbreen, could those data be reproduced here? Following that, it may be worth adding a spatial plot of the precipitation differences using the transects.

4) What is the specific 'simple' 2D interpolation method used in sections 2.3 and 2.5?

5) Is the code used to generate this dataset available?

---

## Author Response (AR1)

**Response to review comments and revision of ESSD-2019-180**

**Sval_Imp: A gridded forcing dataset for climate change impact research on Svalbard**

**by: Thomas Vikhamar Schuler and Torbjørn Ims Østby**

We thank the reviewers for their constructive comments that have been helpful to improve the quality of our manuscript. Below, we respond to the comments point by point and outline where and how we implemented them in the revised version.
To illustrate the flow of discussion, we use different font colours for the comments by the reviewers (black) and our responses (red).

The attached difference manuscript illustrates the changes in the revised text: blue fonts denote new text and text removed is shown in red color.

**Reviewer Kabir Rasouli**
*General comments*
1. The accumulated snow on ice cannot be representative of precipitation as snow cover and depth on ice are not uniform because of blowing wind effect on redistribution of snow, roughness of ice surface, and spatial variability of blowing snow sublimation (Line 11-12 and Fig. 6). This is clearly shown by a poor relationship between unscaled ERA-interim winter precipitation and snow water equivalent on Figure 6. Either clarify this or remove the text and figures related to this argument.

> We agree that redistribution of snow by wind and mass fluxes caused by sublimation/ deposition may impede the comparison of snow accumulating on the ground and precipitation. The redistribution of snow causes considerable spatial variability depending on terrain roughness (e.g. Aas et al., 2016), typically on spatial scales that are below the resolution of our dataset (1 km). However, the measurements used in our comparison (except for those from Hansbreen), are from glaciers that have very gentle surface topography and smooth surfaces (they are not crevassed), hence the influence of redistribution is small (e.g. Taurisano et al., 2007). As described on page 10 line 7-12 (original MS), the rough topography surrounding Hansbreen introduces a cross-glacier variability of snow accumulation that may be responsible for the poor match with downscaled values. Similar mismatch has been found also by others (Aas et al., 2016; Van Pelt et al., 2019).

> Concerning sublimation, Svalbard has high air humidity throughout the year, limiting the potential for sublimation. In an energy-balance study, Østby et al. (2017) estimated sublimation to about 0.016 m w.e yr-1, which is 1-2 orders of magnitude below annual precipitation sums.

> We will extend the discussion section in our manuscript to enhance clarity of our argument (page 10, L180ff of the revised MS)

> We disagree with the reviewer's point about Fig. 6. If snow sublimation were responsible for the poor agreement between measured snow water equivalent and

winter precipitation, the accumulated snow should be smaller than the precipitation. However, the opposite is true and the unscaled ERA precipitation is much smaller than the observed snow water equivalent. Our argument is, that the highly smoothed orography used for the ERA reanalysis is responsible for underestimating the orographic effect on precipitation. When accounting for this in our downscaling method, the winter precipitation comes much closer to the observed snow water equivalent. Nevertheless, we do not rule out that sublimation takes place, but as discussed above, this could not serve as an alternative explanation.

The y-axis in Figure 6 is mislabelled and will be changed to "ERAi precipitation (Oct-Apr)", such that it applies to both datasets displayed. We add a legend to discriminate the two datasets. See below, where we present a new, enhanced version of the Fig.6 (see also new Fig. 6 in the revised MS).

2. It is not clear if monthly data for precipitation were used for downscaling the ERA reanalysis products or daily?

No, all variables have been downscaled at steps of 6 hours. We will modify the text to make this explicit. (P2, L45ff of the revised MS)

If monthly data were used, how reliable is monthly-to6hourly temporal downscaling? If daily data were used, why number of months for precipitation is provided in Table 2?

Comparison between downscaled and measured precipitation (for operational weather stations) has been conducted using monthly values. Precipitation in reanalysis is largely unconstrained and small mismatches in timing of precipitation (<1 day) would penalize a method that otherwise is successful in reducing the underestimation of the reanalysis (at least on monthly and seasonal scale). Table 1 shows the number of values that have been used in the evaluation of different variables. While T2, RH etc have been evaluated using daily values, precipitation has been evaluated using monthly values. We will modify the text and the caption to better explain our procedure (P9, L160 ff in the revised MS). We believe that the reviewer refers to this table and not Table 2 that lists the files in the dataset, organized in monthly chunks.

For other variables if daily variables recorded at the stations were used how were the daily values disintegrated to 6-hourly data?

The observational records have been used to evaluate the downscaled variables. The 6-hours values of the downscaled variables have been aggregated to daily values for the evaluation. In the revised version, we better explain this point (P9, L160ff, in the revised MS).

3. I am wondering how many of the stations that were used for validation of the downscaled meteorological variables were originally used to assimilate the reanalysis products. Is it only the Ny-Ålesund station? If the station data have been used in assimilating reanalysis products, it is not surprising to see high correlations between time series of ERA interim grids and stations.

Presumably, all available data of the met-office operated weather stations is used for constraining the reanalysis, but a detailed examination of which dataset has been used and when, is difficult. The comment by the reviewer is in line with our

argumentation on P11, L30 (original MS) to explain the superior performance at Ny Ålesund. In contrast, records from project stations represent independent measurements (for instance the stations on Etonbreen and Holtedahlfonna).

**Specific comments: (P: Page, L: Line)**

1. I recommend removing the index of "Sval_Imp_v1" form the title and across the manuscript.

> We will follow the advice of the reviewer and refer to the dataset as "Sval_Imp". This has been implemented throughout the revised MS.

2. P 2 L 13: Rewrite "Whereas operational records are biased to low-elevations . . .".

> Operational records are biased to low-elevations around the fringes of the archipelago. Therefore, we stress the hitherto under-used potential of project-based measurements in the interior, high-elevation regions for evaluating atmospheric models. (modified text in abstract, P1, L9-11 of revised MS).

3. P 4 L 1: Replace "In contrast, elevation in our 1×1 km topography peaks 1600 m asl" with "In contrast, the highest elevation in our gridded topography map is 1600 m asl".

> Done.

4. P 4 L 2: Check the grammar in "present" and also in the next line in "and use". Do you mean you have used linear theory?

> Changed to: "…is not represented by…" and
> "We assume that this is the main reason for the poor performance of reanalyzed precipitation. To account for orographic enhancement, we use a linear theory (LT) of orographic precipitation (Smith and Barstad, 2004)." (revised text Sec 2.1 of revised MS).

5. P4 L10: Add ", which was" before "not present in the ERA products".

> Done.

6. P4, L14-15: The authors have mentioned "we modified the TopoSCALE methodology regarding downscaling of direct shortwave radiation and air temperature, as described in the following." However, the following section is about precipitation and not about radiation or temperature. Use more specific section numbers or names.

> Changed to "regarding downscaling of air temperature and shortwave radiation, as described in Sections 2.3 and 2.4". (P4, L73-74 of revised MS).

7. Use "air" temperature instead of temperature when you refer to air temperature. This can be mixed up with surface or soil temperatures if not specified.

> Done.

8. P10 L12: Change "Grabiec et al. (2006)." to "(Grabiec et al., 2006)."

> Done.

9. Is the dataset presented in this paper and stored in the website only downscaled products or both raw unscaled and processed downscaled data (e.g., Table 2)?

> The published dataset contains only the downscaled data. In the data availability section, we refer to the availability of the unscaled reanalyses.

10. Check for the typos and English grammar errors across the manuscript.

We have re-checked the manuscript to our best knowledge, but of course cannot rule out that it is free of typographical or grammatical errors. Any specific comment is appreciated.

**Reviewer Andrew Newman:**

1) Although the authors give citations to other papers that evaluate this product, it would be beneficial to have a more explicit summary of those results in this paper so they are more readily available in one location.

As stated at the onset of Sec 3 (p8, l9-11, original MS), our manuscript summarizes the evaluations presented in previous papers in the subsequent subsections. We make this more transparent by writing: "... in this paper, we summarize their main results in the subsequent subsections, and present additional evaluation of the precipitation using snow measurements." (P9, L159-160 of revised MS).

2) The paper needs more discussion of the differences in performance between the ERA-40 and ERA-Interim time periods using the overlap period wherever possible. This will be very useful to users of the full dataset. There is some discussion of this for temperature, but not for any other variables.

Due to sparsity of available data in the overlap period 1979-2002, we have not evaluated the performance of Sval_Imp for other variables than air temperature.

However, Østby et al. (2017) have evaluated the effect of this dataset discontinuity by simulating glacier mass balance using both, ERA-40 and ERA-Interim to investigate whether this discontinuity could be responsible for a notable drop in simulated mass balance around year 1980.

They found that the ERA40-based simulation yields an about 13 cm w.e. higher mass balance than the ERA-Interim-based one, but ERA-40- based simulations still show a 20 cm drop of mass balance between 1970 and 1990, larger than that caused by the data set discontinuity. This suggests that this change in mass balance regime was not caused by the heterogeneity of our composite forcing. Nevertheless, we cannot rule out the possibility that this change was caused by the discontinuity inherent in both reanalyses due to the availability of satellite observations after 1979. (new text in Sec 3.2 of revised MS).

3) Are there any other snow transect data available that could be added to this analysis? It appears that another paper evaluated this product on another glacier named Hansbreen, could those data be reproduced here? Following that, it may be worth adding a spatial plot of the precipitation differences using the transects.

Østby et al. (2017) evaluated glacier mass balance that has been simulated using the presented dataset to force a coupled surface energy balance – snowpack model. These measurements do not represent spatially "continuous" transects but a collection of 10 point measurements. A map representation of differences would not provide much insight. Nevertheless, we came over an earlier snow transect dataset by Sand et al (2003) that covers several regions across Svalbard. We include this in our revised version to better evaluate the spatial pattern of precipitation.

Here is a revised version of Fig 6 (p11 of revised MS), now including an evaluation of the spatial pattern using the 1999 dataset by Sand et al. (2003).

[Figure]

Figure 6: a) Map view of Sval_Imp precipitation accumulated over Oct 1998 to Mar 1999, overlaid by coloured circles, indicating the measured snow water equivalent (SWE) by Sand et al. (2003). b) scatterplot comparing the 1999 measurements to winter precipitation according to Sval_Imp (crosses) and to ERA-interim (dots). c) similar scatterplot as in b) but only for the Austfonna dataset that provides multi-temporal coverage.

4) What is the specific 'simple' 2D interpolation method used in sections 2.3 and 2.5?

Instead of interpolating a 3D data volume to the surface elevation of the high resolution topography, for some variables, we have bilinearly interpolated the 2D field representing the values at 2 m height. We modify the text to be more explicit in this regard. (P7, L129 ff and P8 L154 ff of revised MS)

5) Is the code used to generate this dataset available?

Unfortunately, the code has been developed to solve the task at hand but not having a wider user community in mind. Nevertheless, it is available on request, but sufficient instructions are required to make it applicable. The code for precipitation downscaling for instance, has been used by Roth et al. (2018). Future work will aim at making the code a more useful tool for others.

*Copyright statement.* Sval_Imp is licensed under the Creative Commons Attribution 4.0 International License (CC-BY 4.0). In essence you are free to copy, distribute, and adapt the work, as long as you attribute the work to its origin and abide by the other license terms.

[revised manuscript text omitted]

---

## Author Response (AR2)

**Re: Manuscript essd-2019-180**

Dear editor,

Thanks for your positive response!
We followed your advice to make our code available.
It was created to solve the tasks at hand and has not been designed to provide a streamlined workflow for other users, and many comment sections may be insider jargon rather than directed to a wider public. Nevertheless, it is publicly available under a GNU GPLv3.0 license at the following repository:
https://github.com/TVSchuler/Sval_Imp_matlab

we have updated the data availability section of our manuscript to now include also the above information, see attached document.
Technically, how do we proceed further to upload the updated version?

Thank you very much for your time in this matter!

Best regards,
Thomas V Schuler

--
Dr. Thomas V. Schuler
Professor
Department of Geosciences
University of Oslo
P.O. Box 1047 Blindern
N-0316 OSLO
Norway

Tel.:     +47 22 85 59 28
email:   t.v.schuler@geo.uio.no